# Learning Compressed Shape-Aware Molecular Representations for Virtual Screening

Robin Winter [1]    Julian Cremer [1]    Djork-Arné Clevert [1]

## Abstract

Virtual screening of billion-scale molecular libraries based on 3D shape similarity remains computationally prohibitive, requiring expensive conformational sampling and alignment, as done by established tools like *ROCS*. Here, we introduce *SAND* (**S**hape-**A**ware **N**eural **D**escriptor), a method that can retrieve shape similar molecules from their 2D graph alone. Our approach makes two key contributions: (1) a rank-preserving contrastive learning framework using differentiable Spearman correlation that results in representations where similarity strongly correlates with 3D molecular shape overlap ($R = 0.86$), and (2) an end-to-end learned quantization-aware training scheme that jointly optimizes the encoder with a two-level IVF-PQ discretization step, achieving approximately $4\times$ better compression than post-hoc quantization at equivalent retrieval quality. We demonstrate that *SAND* enables searching over 10 billion molecules in less than a second on a single GPU node — a speedup of $> 10^8\times$ compared to traditional methods. We release open-source code and trained weights to facilitate adoption.

## 1. Introduction

Virtual Screening (VS) is a well-established technique in Early Drug Discovery to identify novel, potentially active chemical matter by searching large molecule libraries based on chemical similarity to a query (e.g., known active) compound (Lavecchia & Di Giovanni, 2013). While similarity is commonly measured based on topological (2D) fingerprints (Todeschini & Consonni, 2000), shape-based approaches have been shown to recover structurally more diverse molecules that preserve similar pharmacophoric fea-

tures in 3D space (Venhorst et al., 2008; Hawkins et al., 2007). However, computing shape similarity requires expensive conformational sampling and pairwise 3D alignment—costs that scale poorly to modern virtual libraries containing $10^9$–$10^{15}$ compounds (Grygorenko et al., 2020). The computational burden is substantial: a single flexible molecule may require hundreds of conformations to adequately represent its shape space (Hawkins, 2017), and aligning two conformational ensembles scales quadratically. Even GPU-accelerated tools like ROCS/FastROCS (Grant et al., 1996), capable of $10^7$ alignments per second, would require multiple GPU-years to exhaustively screen a multi-billion-compound library against a single query with adequate conformational sampling. This creates a fundamental gap between the potential benefits of shape-based VS and its practical applicability at scale.

To address this challenge, we propose *SAND* (**S**hape-**A**ware **N**eural **D**escriptor), a learned molecular graph representation where cosine similarity directly approximates 3D shape overlap, eliminating conformer generation and alignment at inference time. Our approach makes two main contributions:

- **Rank-preserving contrastive learning.** We introduce a contrastive learning framework that optimizes a differentiable Spearman correlation loss between embedding similarities and ground-truth shape overlap scores. This produces representations in which the nearest-neighbor retrieval recovers shape-similar molecules ($R = 0.86$ correlation with 3D overlay scores).

- **End-to-end quantization-aware training.** We jointly train the encoder with a two-level IVF-PQ discretization step utilizing neural discrete representation learning, achieving $4\times$ better compression than post-hoc k-means quantization at equivalent retrieval quality.

The trained model integrates directly into optimized vector database frameworks such *as FAISS* (Douze et al., 2024), allowing search over more than 10 billion molecules in less than a second on a single GPU node. Beyond accelerating traditional virtual screening, this capability opens a new application: bridging generative chemistry with pur-

[1]Pfizer, Department of Machine Learning Research, Berlin, Germany. Correspondence to: Robin Winter <robin.winter@pfizer.com>.

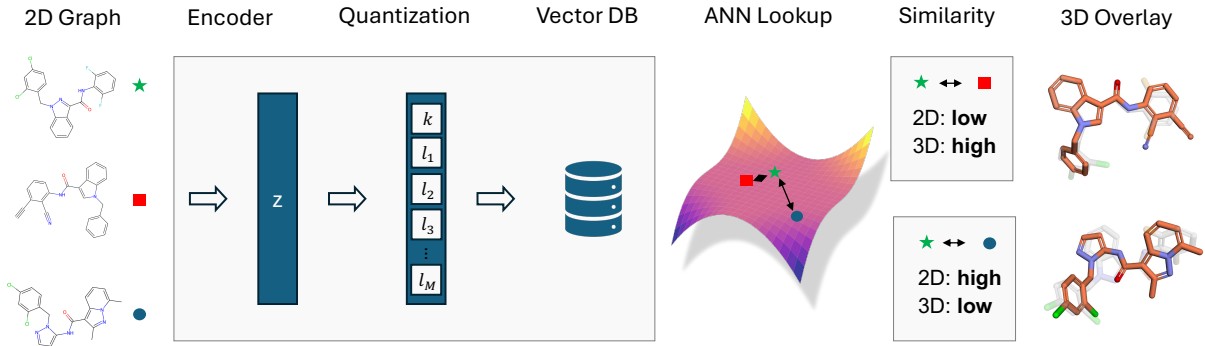

*Figure 1.* Overview of the proposed *SAND* framework. A graph neural network encodes 2D molecular graphs into continuous vector representation, followed by a trained discretization step utilizing an inverted file (IVF) and product quantization (PQ) codebook. With the trained model, large molecular databases can be encoded, quantized and stored in a vector database to enable rapid approximate nearest neighbor (ANN) lookups based on shape similarity.

chasable chemical matter. Deep generative models increasingly produce high-quality ligand candidates, yet many remain synthetically intractable (Gao & Coley, 2020). We demonstrate that *SAND* can retrieve commercially available shape-analogs for AI-generated compounds, providing a practical pathway from *in silico* design to experimental validation.

We provide open source implementations of the model and trained model weights based on the open source shape alignment tool *ROSHAMBO2* (Atwi et al., 2025)[1].

## 2. Background and Related Work

**Virtual screening** Virtual screening methods can be broadly categorized into structure-based and ligand-based approaches (Lavecchia & Di Giovanni, 2013). *Structure-based virtual screening* (SBVS) relies on the 3D structure of a biological target, e.g., a protein binding pocket, to rank potential ligands, typically using molecular docking (Lyne, 2002). *Ligand-based virtual screening* (LBVS) instead assumes that similar molecules have similar properties (Bender & Glen, 2004) and identifies new candidates based on their similarity to known active compounds. Most commonly, similarity is measured by comparing 2D topological graphs using circular or path-based fingerprints (Todeschini & Consonni, 2000). Because such fingerprints do not capture 3D shape information, they may miss candidates that are topologically dissimilar but share similar 3D shapes and interaction profiles (Rush et al., 2005). To address this, LBVS methods based on conformational superposition have been proposed and implemented in various tools (Grant et al., 1996; Lemmen et al., 1998; Yan et al., 2013; Cleves et al., 2019; Bolcato et al., 2022; Atwi et al.,

2025). Methods such as *ROCS* (Grant et al., 1996) maximize volumetric overlap between molecular conformations represented as Gaussian functions over shape and pharmacophoric features. Even with efficient CPU and GPU implementations, the need to sample many conformations per flexible molecule makes large-scale screening computationally infeasible. Recently, virtual on-demand libraries (VLs) like *Enamine REAL Space* (Grygorenko et al., 2020) are rapidly growing in size and cover an increasingly larger ($10^9 - 10^{15}$) chemical space while becoming cheaper to order from ($\sim$ \$100 per compound)—furthering the need for scalable alternatives to traditional shape-based virtual screening methods.

**Machine-Learned Molecular Descriptors** Recently, much work has been done to learn molecular representations rather than relying on handcrafted feature extraction protocols such as circular fingerprints. Common strategies involve unsupervised/self-supervised learning techniques, such as autoencoders, masked-prediction or contrastive learning, to capture the underlying structure of the input molecular representation (e.g., topological graphs or SMILES strings) (Winter et al., 2019; Wang et al., 2019b; Jaeger et al., 2018). These learned representations have been shown to be useful descriptors of molecules and can be utilized in various downstream tasks such as property prediction or virtual screening. Naturally, the structure of the learned latent space is highly dependent on the training objective and data used during training. Although most of the work focuses on learning representations that capture only topological similarity, some works have explored learning representations that capture 3D shape similarity. Methods like 3D-Infomax (Stärk et al., 2022) and GraphMVP (Liu et al., 2021) utilize contrastive learning between 2D graph and 3D conformer views to inject geometric awareness into

---

[1]Code and trained model weights will be made available at https://github.com/pfizer-opensource/SAND.

molecular representations. Other approaches like SchNet (Schütt et al., 2018) and DimeNet (Gasteiger et al., 2020) directly operate on 3D coordinates but require conformer generation at inference time. The methods most closely related to our proposed approach are recent works by Atsango et al. (2022); Sellner et al. (2023); Lžičař & Gamouh. They propose learning shape-aware molecular representations by utilizing contrastive learning based on sets of molecules for which pairwise shape similarities are calculated using traditional shape alignment tools such as *ROCS*. However, these methods either do not address scalability to billion-scale libraries or rely on post-hoc quantization of the learned representations.

**Vector Databases and Approximate Nearest Neighbor Lookups**  With the rise of machine-learned, high-dimensional, continuous vector representations for various data modalities such as images and texts, a new class of databases has emerged to store and query these high-dimensional vectors efficiently (Ma et al., 2023). These *vector databases* are optimized for storing, indexing and searching large collections of vectors, enabling fast similarity searches and nearest neighbor lookups. Commonly, approximate nearest neighbor (ANN) algorithms are used to trade off recovery accuracy to significantly improve query speed and scalability (Li et al., 2020). Popular ANN algorithms include Locality-Sensitive Hashing (LSH) (Gionis et al., 1999), Inverted File Lists (IVF) (Zobel & Moffat, 2006) and hierarchical navigable small world (HNSW) graphs (Malkov & Yashunin, 2018). Storage is usually optimized by utilizing compression techniques such as vector (VQ) (Gray, 1984) or product (PQ) quantization (Jegou et al., 2010). Libraries such as *FAISS* (Douze et al., 2024), provide efficient end-to-end implementations for indexing and querying large vector datasets and optionally provide GPU acceleration. A commonly used structure is the IVF-PQ index, which combines an inverted file index with product quantization to enable efficient and scalable similarity searches in large vector datasets. Inverted file indices partition the vector space into clusters, allowing for fast retrieval of candidate vectors from the nearest clusters for a given query. Product quantization compresses the indexed vectors into sets of integer-based codes, where the corresponding codebooks are usually trained using k-means clustering on a representative subset of the data. This however, introduces a two-stage pipeline with inherent limitations: the encoder must first be fully trained, after which the entire dataset requires encoding and temporary storage in uncompressed form—potentially terabytes for billion-scale libraries—before quantization codebooks can be learned. More fundamentally, encoders trained without quantization awareness might produce representations that are poorly suited for discretization, as the optimization objective provides no incentive to structure the embedding space with balanced subspace variances and

independent dimensions—properties critical for minimizing product quantization distortion (Ge et al., 2013).

**Neural Discrete Representation Learning**  While machine-learned representations are usually continuous, recently methods have been proposed to learn discrete representations using neural networks (Oord et al., 2018; Razavi et al., 2019; Zeghidour et al., 2021). Following the idea of vector quantization (Gray, 1984), these methods aim to learn a finite set of vectors (codebook) to which points of the continuous space are mapped to and reconstructed from. Usually, an encoder-decoder architecture is employed and trained on minimizing the reconstruction error while approximating the gradient of the non-differentiable discretization step with a *straight-through estimator* (Bengio et al., 2013) or rotation trick (Fifty et al., 2025). In the context of large-scale retrieval, prior work has explored learning quantization end-to-end with encoder networks. Deep Visual-Semantic Quantization (Cao et al., 2017) and Product Quantization Networks (Yu et al., 2018) jointly learn PQ codebooks with image encoders for retrieval tasks, while ScaNN (Guo et al., 2020) introduces anisotropic quantization that accounts for retrieval-specific objectives. However, these methods typically focus on single-level quantization schemes. Hierarchical approaches like residual vector quantization, as used in neural audio codecs (Zeghidour et al., 2021; Défossez et al., 2023), demonstrate the benefits of multi-level discretization but optimize for reconstruction rather than retrieval. To the best of our knowledge, no prior work has jointly learned a two-level IVF-PQ index structure end-to-end with a domain-specific encoder optimized for rank correlation with a target similarity metric, while maintaining direct compatibility with production vector database implementations like *FAISS*.

## 3. SAND

### 3.1. Encoder

A main factor that slows down traditional shape-based virtual screening is the dependence on sampling and alignment of 3D conformations. To avoid this, we propose to learn an encoder that maps 2D molecular representations to a continuous vector space that reflects shape similarity. While many options exist, here we utilize the *Graph Neural Network* (GNN) variant *GINE* (Hu et al., 2020) as encoder architecture. This encoder $E$ takes as input a molecular graph $\mathcal{G}(\mathcal{X}, \mathcal{E})$ with node $\mathcal{X}$ and edge $\mathcal{E}$ features and maps it to a $D$-dimensional continuous vector representation $\mathbf{z} \in \mathbb{R}^D$:

$$\tilde{\mathbf{z}} = E(\mathcal{G}(\mathcal{X}, \mathcal{E})) \tag{1}$$

followed by a normalization:

$$\mathbf{z} = \frac{\tilde{\mathbf{z}}}{||\tilde{\mathbf{z}}||} \,. \tag{2}$$

### 3.2. Contrastive Representation Learning

To learn molecular representations that reflect a custom similarity, such as shape overlay, we utilize a contrastive learning approach based on sets of molecules $(\mathcal{M}_q; \mathcal{M}_1, \ldots, \mathcal{M}_i, \ldots, \mathcal{M}_N,)$ for which the relative similarity ranks $r_{q,i}$ between query molecule $\mathcal{M}_q$ and molecules $\mathcal{M}_i$ are known. Our goal is to train the encoder such that the similarities $s_{q,i}$ between molecular representations of query $\mathbf{z}_q$ and matches $\mathbf{z}_i$ are correlated to their similarity ranks $r_{q,i}$. To achieve this, we utilize a differentiable approximation of the *Spearman's rank correlation coefficient* $\hat{R}(\cdot, \cdot)$ (Blondel et al., 2020), optimizing the loss:

$$\begin{aligned}
\mathcal{L}_{\text{corr}} &= -\hat{R}[\mathbf{s}, \mathbf{r}] \\
&= -\hat{R}[(s_{q,0}, \ldots, s_{q,N}), (r_{q,0}, \ldots, r_{q,N})]
\end{aligned} \tag{3}$$

In our model, we chose the similarity measure $s_{q,i}$ to be the dot-product between the normalized molecular representations:

$$s_{q,i} = \mathbf{z}_q \cdot \mathbf{z}_i \,. \tag{4}$$

To encourage utilization of the full representation space on the hypersphere, we further add a *uniformity loss*:

$$\mathcal{L}_{\text{uni}} = \frac{1}{(N-1)^2} \sum_{i=1}^{N} \sum_{\substack{j=1 \\ j \neq i}}^{N} \min(\alpha, \mathbf{z}_q^i \cdot \mathbf{z}_q^j) \,, \tag{5}$$

where $\alpha$ is a margin hyperparameter, pushing query molecule representations in a mini-batch away from each other.

### 3.3. Quantization

To enable efficient storage and querying of the learned molecular representations, we utilize a neural discrete representation learning approach (Oord et al., 2018) to learn both an inverted file (IVF) and product quantization (PQ) codebook in an end-to-end fashion. The IVF codebook is comprised of $K$ code vectors $\mathbf{c}_{\text{ivf}} \in \mathbb{R}^D$, which partition $\mathbf{z}$ into $K$ clusters. The PQ codebook further compresses the residuals $\mathbf{r} = \mathbf{z} - \mathbf{c}_{\text{ivf}}^k$ by splitting $\mathbf{r}$ into $M$ sub-vectors and quantizing each sub-vector with its own codebook containing $L$ code vectors $\mathbf{c}_{\text{pq}}^l \in \mathbb{R}^{D/M}$. The final discretized representation $\mathbf{q}$ of a molecule is thus comprised of the IVF code index $k$ and $M$ PQ code indices $l_m$:

$$\mathbf{q} = (k, l_1, \ldots, l_{M-1}, l_M) \,.$$

To reconstruct $\mathbf{z}$ from $\mathbf{q}$, the assigned IVF code vector is added to the concatenated PQ code vectors:

$$\hat{\mathbf{z}} = \mathbf{c}_{\text{ivf}}^k + \bigoplus_{m=1}^{M} \mathbf{c}_{\text{pq}}^{l_m} \,, \tag{6}$$

where $\bigoplus$ denotes the concatenation operation. Following Oord et al. (2018), we use exponential moving average (EMA) updates to learn the codebook vectors. For each code vector $\mathbf{c}_i$, we maintain a count $N_i$ and an accumulated sum $\mathbf{m}_i$ of the encoder outputs assigned to it, updated as:

$$N_i^{(t)} = \gamma N_i^{(t-1)} + (1 - \gamma) n_i^{(t)} \tag{7}$$

$$\mathbf{m}_i^{(t)} = \gamma \mathbf{m}_i^{(t-1)} + (1 - \gamma) \sum_j \mathbf{x}_j^{(t)} \tag{8}$$

where $\gamma$ is the decay rate, $n_i^{(t)}$ is the number of encoder outputs assigned to code $i$ in the current batch, and the sum is over all encoder outputs $\mathbf{x}_j$ assigned to code $i$. The code vectors are then updated as

$$\mathbf{c}_i^{(t)} = \mathbf{m}_i^{(t)} / N_i^{(t)} \,. \tag{9}$$

To encourage the encoder to learn representations that correlate well with the similarity metric even after quantization, we apply the loss in Eq. 3 to the reconstructed vectors $\hat{\mathbf{z}}$. A *straight-through estimator* (Bengio et al., 2013) is used to approximate the gradients of the non-differentiable quantization step, copying gradients from the quantizer's output directly to the quantizer's inputs. As proposed by Oord et al. (2018), the quantizer's inputs are encouraged to stay close to the codebook vectors, by adding a commitment loss:

$$\mathcal{L}_{\text{com}} = \beta \cdot \mathcal{D}(\mathbf{x}, \mathbf{c}), \tag{10}$$

where $\mathbf{x}$ denotes the input to the quantization step, $\mathcal{D}$ is a distance measure and $\beta$ is a hyperparameter controlling the strength of the commitment loss.
We choose $\mathcal{D}^{\text{IVF}}(\mathbf{x}, \mathbf{c}) = \mathbf{x} \cdot \mathbf{c}$ and $\mathcal{D}^{\text{PQ}}(\mathbf{x}, \mathbf{c}) = ||\mathbf{x} - \mathbf{c}||_2^2$, for the IVF and PQ quantization steps respectively.

### 3.4. Training Objective and Inference

The final training objective is a weighted combination of the two contrastive losses $\mathcal{L}_{\text{corr}}, \mathcal{L}_{\text{uni}}$ and the two commitment losses $\mathcal{L}_{\text{com}}^{\text{IVF}}, \mathcal{L}_{\text{com}}^{\text{PQ}}$:

$$\mathcal{L} = \mathcal{L}_{\text{corr}} + \gamma \mathcal{L}_{\text{uni}} + \beta^{\text{IVF}} \mathcal{L}_{\text{com}}^{\text{IVF}} + \beta^{\text{PQ}} \mathcal{L}_{\text{com}}^{\text{PQ}} \,. \tag{11}$$

After training, the learned IVF and PQ codebooks can directly be integrated in an inverted file list plus product quantization (IVF-PQ) index structure as implemented in *FAISS* (Douze et al., 2024). A large database of molecules can be stored and indexed by passing them through the models encoder, followed by *FAISS* quantization and storage, without the need for any further training of the *FAISS* index.

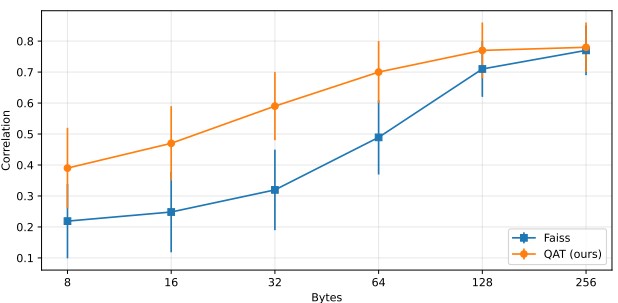 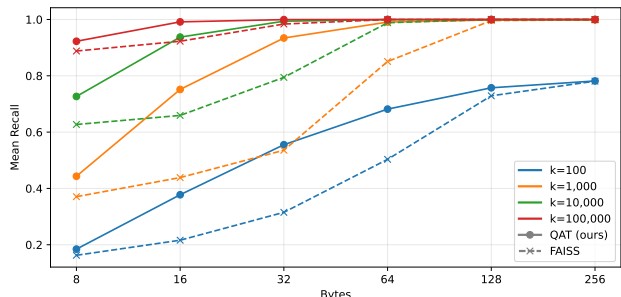

*Figure 2.* Compression performance comparison between our quantization aware training (QAT) method and a FAISS IndexIVFPQ baseline. The x-Axis shows the number of Bytes used to compress the residuals, which corresponds to the number of (1 Byte) PQ subquantizers $M$ used. **Left**: Mean Spearman's rank correlation between the quantized embedding cosine similarities and ground-truth Morgan FP similarities. **Right**: Mean Recall@k of recovering the top 100 most similar molecules based on Morgan FP similarities.

## 4. Experiments

To evaluate the proposed method, we focus on two main questions: (i) how effectively the learned representations can be compressed while preserving similarity structure, and (ii) how accurately shape similarity can be approximated from 2D molecular graphs to enable rapid shape-based virtual screening at virtual-library scale.

### 4.1. Compression Performance

To evaluate the compression performance, we train models with varying compression rates and evaluate the recovery rates and downstream correlation on a hold-out test set. For this experiment, we utilize the diverse $1\%$ subset of *Enamine REAL Database* (Grygorenko et al., 2020) with approximately 10 million molecules for training. We calculated Morgan Fingerprint (FP) similarities between all molecule pairs to find the top 1000 most similar molecules based on Tanimoto similarity[2]. We train models with varying compression rates by changing the number of PQ sub-vectors $M$ while keeping the PQ codebook sizes $L = 256$ (1 Byte) constant. The IVF codebook size is fixed to $K = 1024$. After training, we initialize *FAISS's* IVF-PQ index with the extracted IVF and PQ codebooks of the trained model. As baseline, we train another model with the same hyperparameters but without the quantization step. Afterwards, we extract the embeddings of the training data and utilizes *FAISS's* build-in k-means clustering to train the codebooks post-hoc. As test set, we extract a random subset of 1 million molecules from the *Enamine REAL Database* and calculate the full pairwise Morgan FP tanimoto similarity matrix.

As shown in Figure 2, our method significantly outperforms the baseline across all compression rates, indicating that learning the codebooks in an end-to-end, quantization-aware

fashion leads to more effective compression of the molecular representations with respect to the similarity metric of interest. For example, a quantization-aware training with a $M = 16$ compression leads to a similar downstream correlation as training a $M = 64$ *FAISS* compression on quantization-unaware model, i.e., leading to an approximately four times higher compression performance.

As a sanity-check, we retrained the codebooks of the quantization-aware model, extracting embeddings before quantization, using *FAISS*. This resulted in a similar quantization performance, indicating that the learned representations are more amenable to quantization in general. This result can be further quantified by investigating the geometric complexity of the learned representation. We compared the quantization-aware model ($m = 32$) against a model trained without quantization using otherwise the same hyperparameters. For a set of 1 million molecules from the test set, we extracted the embeddings and analyzed their covariance spectrum, calculated the number of principal components (PC) required to explain $> 90\%$ of the variance. While the quantization-unaware model required 317 PC, the quantization-aware model only required 83 PC, i.e., a markedly steeper variance spectrum, indicating a lower intrinsic dimensionality and explaining the improved compression performance.

### 4.2. Shape Overlay Score Approximation

Next, we evaluate how accurately shape similarity can be approximated from 2D molecular graphs. For this experiment, we utilized the same data from Enamine as in Experiment 4.1, but calculate shape similarities between molecule pairs. Our proposed method is agnostic to the choice of shape similarity calculation method, however, since *OpenEye's ROCS* license prohibits open-sourcing weights of models trained on results of their software, we utilize the open-source shape alignment tool *ROSHAMBO2* (Atwi et al., 2025) in the following experiments. For the final training set, we sample up to 300 conformations per molecule, using *RDKit's*

---

[2]This experiment investigates the compression performance under an arbitrary similarity metric. We utilize molecular FP similarities here, since they can be acquired with significantly less computational cost compared to shape similarities.

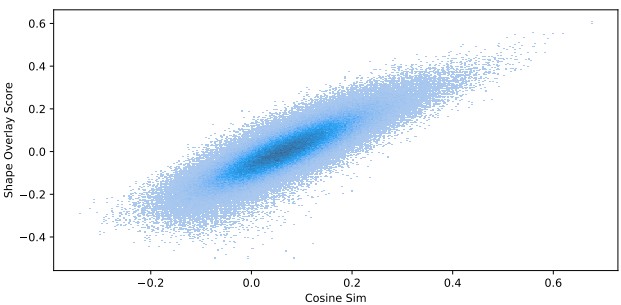 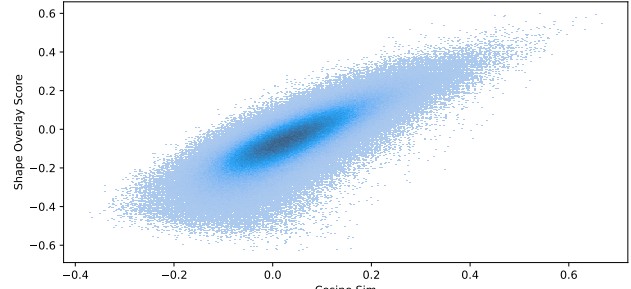

*Figure 3.* Shape overlay scores vs cosine similarities of the learned *SAND* representations between query molecules (**Left**: 500 Enamine compounds, $R = 0.86$; **Right**: 1000 ChEMBL, $R = 0.81$) and best 1000 matches for each query in a fixed reference set from Enamine.

*ETKDGv3* algorithm (Landrum; Riniker & Landrum, 2015), pruning duplicated conformations with a root-mean-square deviation (RMSD) $< 0.5$ Å, followed by an energy optimization (Halgren, 1996). For each molecule pair, we calculate the maximum combined shape and color Tanimoto similarity across all conformation pairs. We utilized the same diverse $1\%$ subset of *Enamine REAL Database* as in Experiment 4.1 for training. As the overlay score calculations are computationally much more expensive, we only calculate shape similarities for 200,000 random query molecules against the top 2048 most similar molecules based on a pre-screening done by an early version of our method, trained on a smaller subset of molecules with less conformations per molecule.

After training, we evaluate the correlation between the learned *SAND* representations and the calculated shape similarities on two held-out query test sets: (i) a random set of 500 unseen molecules from *Enamine REAL Database*, and (ii) a random set of 1000 molecules from *ChEMBL* (Gaulton et al., 2017). As reference set, we utilize a fixed set of 100,000 random unseen molecules from *Enamine REAL Database*. This limited reference set may not include the globally best matches, but allows us to calculate the full pairwise shape similarity matrix between all query and reference molecules with up to 300 conformations in a reasonable time.

In Figure 3, we show the correlation between the maximum shape overlay scores and cosine similarities of the learned *SAND* representations (512 dimensional) between query molecules and their best 1000 matches against the fixed reference set from *Enamine*. We observe a strong Spearman correlation between similarities of the learned representations and the actual shape similarities of 0.86 and 0.81 for Enamine and ChEMBL, respectively. In Table 1 we compare SAND based top-k retrieval recall to baseline methods such as simple 2D Extended Connectivity Fingerprints (ECFP) (Rogers & Hahn, 2010) and pretrained 3D-aware molecular representations, namely 3DInfomax (Stärk et al., 2022), GraphMVP (Liu et al., 2021) and DrugCLIP (Gao et al., 2023)). While GraphMVP ($R = 0.37$) and Drug-

CLIP ($R = 0.31$) achieve higher *global* correlation with shape scores than ECFP ($R = 0.20$), they all underperform ECFP on top-100 recall. This is probably because top shape matches often share common substructures with the query, making ECFP a competitive baseline for top-$k$ retrieval despite its lack of explicit 3D awareness. All baselines remain far below *SAND* ($R = 0.81$–$0.86$), which is trained directly on shape overlay scores.

Although slightly lower, the correlation on the ChEMBL test set indicates that the learned representations even generalizes to a previously unseen chemical space. When training with $M = 32$ quantization, corresponding to a $64\times$ compression rate, the correlation decreases by approximately $9\%$ ($R = 0.78$ on Enamine, $R = 0.73$ on ChEMBL).

To demonstrate the utility of the learned representations for shape-based virtual screening, we encoded the approximately 10 million molecules from the $1\%$ *Enamine REAL Database* subset in a FAISS index using our trained $M = 32$ model's encoder and codebooks. In Figure 4, we show for an example query from ChEMBL the top 1000 matches retrieved by *SAND* together with those retrieved by Morgan FP similarity. While there is some overlap between the two methods, most of the top FP matches have comparably low shape similarity to the query, returning molecules with the same local chemotypes but arranged in a different global 3D shape (Molecules E,F). In contrast, *SAND* matches are more enriched with high shape similar compounds, including molecules with different chemotypes but similar overall 3D shape (Molecules B-D), demonstrating the advantage of our proposed method.

## 4.3. Encoder, loss, and quantizer ablations.

To isolate the contribution of individual design choices, we trained variants of *SAND* that swap (i) the GNN backbone (GINE vs. GPS (Rampášek et al., 2022) vs. GATv2 (Brody et al., 2022)), (ii) the loss function (Spearman vs. triplet vs. MSE), and (iii) the quantizer (IVF-PQ vs. residual vector quantization, RVQ (Babenko & Lempitsky, 2014)) (cf. Table 1). Across both Enamine and ChEMBL test sets, GPS

*Table 1.* Recall@$k$ for shape overlay retrieval of the top 100 most similar molecules ($k = 10^x$). SAND is compared against the 2D fingerprint baseline (ECFP, radius 2, 2048 bits), three pretrained 3D-aware molecular representations (3DInfomax (Stärk et al., 2022), GraphMVP (Liu et al., 2021), DrugCLIP (Gao et al., 2023)), and encoder/loss ablations (GPS, GATv2, Triplet, MSE). Encoder ablations (GPS, GATv2) use the Spearman loss; loss ablations (Triplet, MSE) use the GINE backbone.

| | | Baselines | | | | Ablations | | | |
|---|---|---|---|---|---|---|---|---|---|
| $x$ | SAND | ECFP | 3DInfomax | GraphMVP | DrugCLIP | GPS | GATv2 | Triplet | MSE |
| | | | | *Enamine* | | | | | |
| 2 | $0.50 \pm 0.10$ | $0.13 \pm 0.06$ | $0.03 \pm 0.02$ | $0.05 \pm 0.03$ | $0.04 \pm 0.03$ | $0.51 \pm 0.11$ | $0.50 \pm 0.10$ | $0.42 \pm 0.13$ | $0.46 \pm 0.10$ |
| 3 | $0.89 \pm 0.08$ | $0.30 \pm 0.11$ | $0.10 \pm 0.07$ | $0.16 \pm 0.10$ | $0.14 \pm 0.10$ | $0.91 \pm 0.09$ | $0.89 \pm 0.08$ | $0.82 \pm 0.12$ | $0.88 \pm 0.09$ |
| 4 | $0.99 \pm 0.01$ | $0.60 \pm 0.13$ | $0.34 \pm 0.13$ | $0.48 \pm 0.17$ | $0.46 \pm 0.17$ | $0.99 \pm 0.01$ | $0.99 \pm 0.01$ | $0.98 \pm 0.02$ | $0.99 \pm 0.01$ |
| | | | | *ChEMBL* | | | | | |
| 2 | $0.34 \pm 0.15$ | $0.05 \pm 0.04$ | $0.01 \pm 0.01$ | $0.02 \pm 0.02$ | $0.02 \pm 0.03$ | $0.39 \pm 0.11$ | $0.33 \pm 0.12$ | $0.28 \pm 0.11$ | $0.31 \pm 0.12$ |
| 3 | $0.75 \pm 0.19$ | $0.15 \pm 0.09$ | $0.04 \pm 0.04$ | $0.10 \pm 0.07$ | $0.10 \pm 0.08$ | $0.81 \pm 0.10$ | $0.74 \pm 0.11$ | $0.66 \pm 0.09$ | $0.71 \pm 0.12$ |
| 4 | $0.97 \pm 0.07$ | $0.42 \pm 0.14$ | $0.21 \pm 0.11$ | $0.42 \pm 0.17$ | $0.36 \pm 0.17$ | $0.98 \pm 0.07$ | $0.97 \pm 0.08$ | $0.93 \pm 0.07$ | $0.95 \pm 0.09$ |

yields only marginal gains over GINE despite a $3\times$ larger parameter count (87M vs. 28M), confirming that *SAND*'s gains stem from the training procedure rather than the encoder backbone. The Spearman loss consistently outperforms triplet and MSE alternatives by leveraging batch-wide ranking information from comparable shape overlay scores. While RVQ achieves higher overall correlation (Enamine $R = 0.84$ vs. 0.78; ChEMBL $R = 0.81$ vs. 0.73), it matches IVF-PQ on top-100 recall. As its iterative beam-search encoding is significantly slower at query time than the highly optimized IVF-PQ GPU kernels in *FAISS* (Douze et al., 2024), for billion-scale latency-critical retrieval, IVF-PQ remains arguably the more practical choice.

## 4.4. Large-scale Shape-based Virtual Screening

Next, we used the trained model with $M = 32$ quantization to encode the whole *Enamine REAL Database* with approximately 10 billion Molecules and store them as a *FAISS* IVPQ index. In total, the index size is only approximately 350 GB. Depending on the available hardware, the index can either be searched with a single CPU as on-disk index (reducing the memory footprint), distributed across multiple CPU nodes or fully loaded on a GPU node for maximum performance, resulting in query times as low as a few milliseconds per query SMILES (see Table 2), corresponding to speedups of more than $10^8\times$ compared to *FastROCS*. To further boost the virtual screening performance and also

*Table 2.* Query time comparison for searching 10B molecules with *SAND*, *ROCS* (CPU), FastROCS (GPU) and *ROSHAMBO2* (GPU), searching the closest $n_{\mathrm{probe}} = 128$ clusters and returning top 128 matches per query, parallelized over 10,000 queries.

| Configuration | SAND | ROCS | ROSHAMBO2 |
|---|---|---|---|
| 1 CPU (on-disk) | 1.2 s | $\sim 10^{11}$ s | *n/a* |
| 16 CPUs (in-memory) | 0.065 s | *n/a* | *n/a* |
| 8 H100 GPUs | 0.008 s | $\sim 10^6$ s | $\sim 10^7$ s |

overcome a limiting factor of *FAISS's* GPU implementation, which only returns 2048 matches per query at maximum, we can perform the searches in two steps: First, create multiple noisy versions of the encoded queries by adding Gaussian noise ($\sigma = 0.05$) and return the top 2048 matches for each noised query. In a second step, encode and re-rank these matches with the more accurate, unquantized encoder representations. This procedure enabled us to retrieve up to 10,000 high-quality matches for a given query in approximately 5-10 seconds only. As final step, one can optionally re-rank the top k retrieved matches (trading off recall) with the true shape overlay scores, such as *ROCS* or *ROSHAMBO2*. This virtual screening funnel can be used to identify shape-similar compounds with ground-truth-level accuracy from ultra-large virtual libraries in a matter of minutes.

## 4.5. Integration with Generative Chemistry for Rapid Hit Expansion

To demonstrate *SAND*'s utility in early-stage drug discovery, we coupled it with *Flowr* (Cremer et al., 2025), a flow matching-based generative model for structure-aware ligand design. While deep generative models have shown considerable promise in producing ligands that complement protein binding pockets (Schneuing et al., 2024; Guan et al., 2023; Cremer et al., 2025), a persistent translational gap exists between computationally generated molecules and experimentally accessible chemical matter, as many generated structures remain synthetically intractable or prohibitively expensive (Gao & Coley, 2020). This disconnect is particularly problematic in early-stage hit identification campaigns, where rapid, cost-effective library design is essential prior to resource-intensive hit-to-lead optimization. *SAND* directly addresses this bottleneck by enabling instant retrieval of purchasable shape-analogs for any generated compound. To validate this approach across distinct binding sites, we selected two MAPK family kinases as test

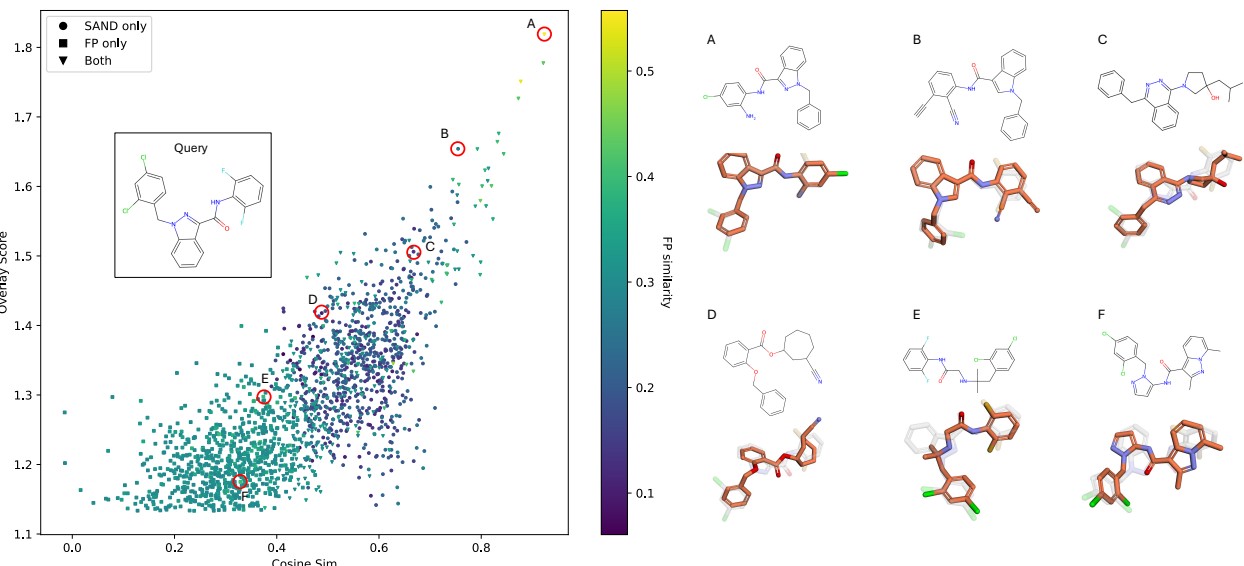

*Figure 4.* **Left**: *Roshambo* shape overlay score over *SAND* representation similarity of top 1000 matches for example ChEMBL query retrieved based on *SAND* and Morgan FP similarity. **Right**: Example matches with best overlay on query. Molecule B,C and D are only in top *SAND* matches. Molecule E and F are only in top FP matches.

cases: c-Jun N-terminal kinase 3 (JNK3/MAPK10) and p38 MAPK. JNK3 is a neuron-specific kinase implicated in neurodegeneration and thus is an attractive therapeutic target for Alzheimer's and Parkinson's disease (Hunot et al., 2004; Koch et al., 2015). p38 MAPK plays a central role in inflammatory signaling and has been pursued as a target for anti-inflammatory therapeutics (Cuenda & Rousseau, 2007; Cicenas et al., 2017). Using *Flowr* conditioned on crystal structures of each target, we generated novel ligand candidates and used representative molecules as queries for *SAND*-mediated analog retrieval. Following Section 4.4, we retrieved the top 1000 shape-matched compounds from the *Enamine REAL Database* for each query. As baseline, we also retrieved the top 1000 ECFP-based analogs and sampled 1000 compounds randomly. The three compound sets were docked into the respective binding sites using Open-Eye Hybrid docking (McGann, 2011; Kelley et al., 2015). To assess pose quality beyond initial docking scores, we performed in-pocket force-field optimization using Open-Eye's *Szybki* with the *ff14SB* force field and calculated the RMSD between the optimized and initially docked poses. Large post-optimization RMSD values indicate conformational instability and suggest poorly fitted binding modes, whereas low values indicate that the docked pose represents a stable energy minimum within the binding site. As shown in the top panels of Figure 5, *SAND*-matched compounds consistently outperformed baseline compounds for both JNK3 and p38 MAPK. The *SAND*-retrieved sets exhibited substantially more favorable Chemgauss4 scores, indicating stronger predicted binding interactions. Criti-

cally, *SAND*-matched compounds also displayed significantly lower post-optimization RMSD values compared to baseline compounds, demonstrating that shape-based retrieval identifies analogs not only achieve favorable docking scores but also adopt more stable binding conformations. For JNK3, we further validated binding potential through MM-PBSA calculations, estimating binding free energies via molecular mechanics energies combined with Poisson-Boltzmann electrostatic solvation (Wang et al., 2019a; Genheden & Ryde, 2015). As shown in the bottom-left panel of Figure 5, the *SAND*-retrieved compounds with a mean score of -32.69 kcal/mol (top-10: -36.24 kcal/mol) exhibit binding energies comparable to or more favorable than the *Flowr*-generated reference (-34.03 kcal/mol).

For p38 MAPK, we examined protein-ligand interaction patterns of three *SAND*-retrieved analogs (Figure 5, bottom right), where interaction diagrams reveal conservation of key binding determinants characteristic of ATP-competitive kinase inhibitors. All three analogs maintain hydrogen bonding interactions with backbone and side-chain residues in the hinge region, complemented by cation-$\pi$ interactions.

Collectively, these results across two kinase targets demonstrate that *SAND* can effectively bridge generative chemistry and experimental tractability. By enabling rapid identification of commercially available compounds that preserve both stable binding conformations and favorable interaction profiles of computationally designed ligands, *SAND* provides a practical pathway for translating generative model outputs into purchasable chemical matter for hit expansion campaigns.

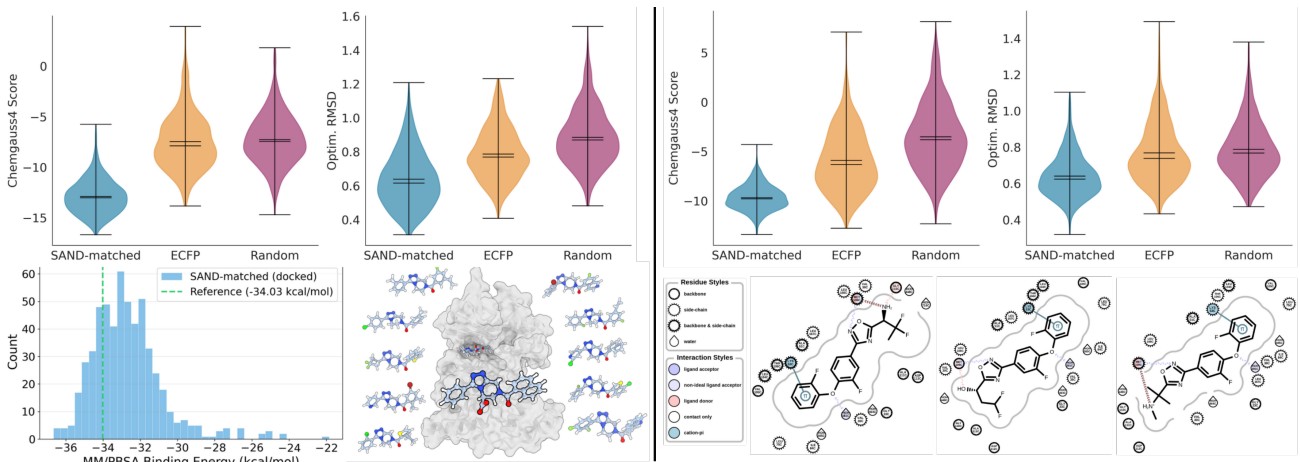

*Figure 5. SAND*-mediated retrieval of purchasable analogs for *Flowr*-generated ligands across two kinase targets. **Top row:** Violin plots comparing docking performance of *SAND*-matched Enamine compounds (blue) versus top ECFP matches (yellow, n=1000) and randomly sampled controls (magenta, n=1000) for *Flowr*-generated ligands targeting JNK3 (left) and p38 MAPK (right). Chemgauss4 docking scores (more negative indicates stronger predicted binding) and post-optimization RMSD values (lower indicates greater pose stability upon force-field refinement) are shown. **Bottom left:** Distribution of MM-PBSA binding energies for *SAND*-retrieved JNK3-targeting compounds. The dashed green line indicates the binding energy of the *Flowr*-generated reference ligand (-34.03 kcal/mol). **Bottom center:** The *Flowr*-generated JNK3 query molecule (blue) positioned within the binding pocket (PDB: 8VS0), surrounded by representative shape-similar Enamine analogs retrieved by *SAND*. **Bottom right:** 2D protein-ligand interaction diagrams for three *SAND*-retrieved analogs docked into the p38 MAPK binding site, illustrating conserved binding interactions including hydrogen bonds to backbone and side-chain residues, with cation-$\pi$ contacts.

## 5. Conclusion

In this work, we proposed *SAND*, a novel framework for rapid shape-based virtual screening of large virtual libraries. We demonstrated how the proposed method can effectively compress molecular representations while preserving similarity structure by jointly optimizing the encoder and quantization codebooks. Moreover, we showed that shape similarity can be accurately approximated from 2D molecular graphs alone. The trained model can be directly integrated in highly optimized vector database frameworks like *FAISS*, allowing for searching large virtual libraries like *Enamine REAL* with approximately 10 billion molecules for shape similar compounds in a few milliseconds. Finally, we demonstrate how *SAND* can be integrated in a generative chemistry workflow, enabling rapid hit expansion of AI-generated compounds to commercially available chemical matter. We release open-source implementations and trained model weights, envisioning *SAND* as a practical tool for accelerating early-stage drug discovery campaigns.

## Impact Statement

This work introduces a novel representation-learning method that accelerates 3D-shape-based virtual screening of ultra-large molecule libraries by several orders of magnitude. This could help accelerated the early-stage drug discovery, making it more efficient and cost-effective to identify promising drug candidates.

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

## A. Training Hyperparameters

Table 3 summarizes all hyperparameters used to train the released *SAND* model. The model was trained on $4\times$NVIDIA H100 GPUs for approximately one week.

*Table 3.* Hyperparameters of the released *SAND* model.

| Group | Hyperparameter | Value |
|---|---|---|
| Encoder (GINE) | MLP layers per GINE block | 3 |
| | Hidden dimension | 1024 |
| | Number of GNN layers | 7 |
| | Output (embedding) dimension $D$ | 512 |
| Loss weights | Uniformity weight $\gamma$ | 0.5 |
| | Uniformity margin $\alpha$ | 0.2 |
| | IVF commitment $\beta^{\text{IVF}}$ | 0.5 |
| | PQ commitment $\beta^{\text{PQ}}$ | 0.25 |
| Optimization | Optimizer | AdamW |
| | Weight decay | 0.05 |
| | Schedule | Cosine annealing w/ linear warmup |
| | Peak / min learning rate | $5\times10^{-4}$ / $1\times10^{-5}$ |
| | Warmup iterations | 10 000 |
| | Decay iterations | 1 000 000 |
| | Batch size | 1024 |
| Compute | Hardware | $4\times$ NVIDIA H100 |
| | Wall-clock training time | $\sim$1 week |

## B. Training Data Bias Analysis

A potential concern with our training-data construction might be that pairwise shape scores are computed only between each query and its top-2048 candidates pre-screened by an early-version *SAND* model, which potentially could introduce bias. The resulting score distribution approximately follows $\mathcal{N}(\mu = 0.30, \sigma = 0.19)$, so the model predominantly trains on moderate-similarity pairs rather than only top hits. To quantify the potential impact of pre-screening, we retrained *SAND* after dropping all training pairs with score $> 0.5$ ($\sim$15% of data) or $< 0.0$ ($\sim$15% of data). This mimics the two main potential issues: (i) prescreening misses the best hits or (ii), prescreening only includes the best hits, which could lead to bad performance on top or low ranks respectively. Results in Table 4 show that performance is largely unaffected, demonstrating that pre-screening does not introduce harmful bias—in particular, removing all pairs with score $> 0.5$ has virtually no impact on top-100 recall, indicating that the model generalizes from learning shape comparisons in the moderate-similarity range. We further note that the test evaluation is fully independent of *SAND*'s training pipeline: test queries are compared against a fixed reference set (100k molecules) using exhaustive pairwise shape overlay—no *SAND* pre-screening is involved.

*Table 4.* Training-data-bias ablation. Recall@$k$ of the top-100 shape matches ($k = 10^x$) when retraining *SAND* after dropping high-score ($> 0.5$) or low-score ($< 0.0$) training pairs.

| Dataset | $x$ | All Scores | Scores $< 0.5$ | Scores $> 0.0$ |
|---|---|---|---|---|
| ChEMBL | 2 | $0.34 \pm 0.15$ | $0.33 \pm 0.15$ | $0.32 \pm 0.16$ |
| | 3 | $0.75 \pm 0.19$ | $0.75 \pm 0.19$ | $0.72 \pm 0.19$ |
| | 4 | $0.97 \pm 0.07$ | $0.97 \pm 0.07$ | $0.96 \pm 0.07$ |
| Enamine | 2 | $0.50 \pm 0.10$ | $0.50 \pm 0.10$ | $0.49 \pm 0.10$ |
| | 3 | $0.89 \pm 0.08$ | $0.89 \pm 0.08$ | $0.88 \pm 0.08$ |
| | 4 | $0.99 \pm 0.01$ | $0.99 \pm 0.01$ | $0.99 \pm 0.01$ |

