# OpenReview forum: "Learning Compressed Shape-Aware Molecular Representations for Virtual Screening"
_ICML.cc/2026/Conference — ICML 2026 regular_

### Official Review · Reviewer_95Jb · 2026-03-10

**Soundness:** 3
**Presentation:** 3
**Significance:** 3
**Originality:** 3
**Overall Recommendation:** 4
**Confidence:** 3

**Summary:**

This paper proposes SAND, an algorithm that learns shape-aware embeddings directly from 2D molecular graphs and reformulates virtual screening as a large-scale vector retrieval problem. The paper is supported by a fairly comprehensive experimental evaluation, including several interesting case studies.

**Compliance With Llm Reviewing Policy:**

Affirmed.

**Final Justification:**

My initial score is already quite high, so I’ve decided to keep my score.

**Key Questions For Authors:**

1. Would the advantage still hold against stronger shape-aware baselines? Since the related work already mentions several methods for learning 3D / shape-aware descriptors, why is the main experimental comparison still primarily against ECFP? It would strengthen the paper to include more systematic comparisons with these more directly relevant baselines.

2. Why was GINE chosen instead of stronger 3D-aware or pretrained molecular encoders? The main contribution of the paper seems to lie more in the retrieval objective and quantization strategy, while the encoder itself is relatively standard. Have the authors considered using a stronger backbone, or combining their framework with existing 3D-aware or pretrained molecular representations?

**Strengths And Weaknesses:**

Strengths

1. The paper is built on a strong and meaningful insight. Traditional 3D shape-based virtual screening requires extensive conformer sampling and pairwise alignment, making it difficult to scale to ultra-large molecular libraries. In this regard, learning a shape-aware representation directly from 2D molecular graphs and turning the problem into vector retrieval is a meaningful direction with potential practical impact.

2. The overall pipeline is fairly complete and well designed. In particular, using differentiable Spearman correlation to directly optimize retrieval ranking consistency, and incorporating quantization into end-to-end training rather than treating it as a post-processing step, are both well aligned with the retrieval setting.

3. The experimental evaluation is relatively comprehensive. The case studies are also interesting and suggest some practical value.

Weaknesses

1. From a technical standpoint, the main components,GINE encoder, IVF-PQ, and FAISS, are not new by themselves. As a result, the paper feels more like a strong systems integration effort around an important problem than a substantial algorithmic advance. The main novelty seems to lie more in the way these components are combined and engineered than in a fundamentally new method.

2. The baselines are somewhat limited. Although the related work discusses a number of prior approaches on 3D / shape-aware descriptor learning, the main retrieval experiments primarily compare against ECFP. This raises the question of whether the claimed advantages would still hold when compared with stronger learned shape-aware baselines.

3. The construction of the training data may introduce selection bias. Instead of computing shape similarity exhaustively over the full space, the paper only performs exact shape calculations for 200,000 queries and the top-2048 candidates pre-screened by an early-version model.

---

> ### Author Rebuttal · Authors · 2026-03-31
>
> **Novelty Beyond Systems Integration.** We appreciate this observation and partially
> agree that the individual components (GINE, IVF-PQ, FAISS) are known. However, we argue
> the contribution extends beyond integration. To the best of our knowledge, no prior work
> has jointly learned a two-level IVF-PQ index structure end-to-end with a domain-specific
> encoder optimized for rank correlation with a target similarity metric while maintaining
> direct FAISS compatibility. Critically, the quantization-aware training is not merely a compression step—it actively reshapes the learned representation geometry to exhibit balanced subspace variances and reduced inter-dimension correlations, properties essential for minimizing PQ distortion that a standard contrastive encoder does not produce (cf. Figure 2: 4× better compression at equivalent recall). Moreover, the specific
> application to shape-aware molecular retrieval with rigorous evaluation on large-scale
> benchmarks represents a novel contribution to the field.
>
> **Stronger Shape-Aware Baselines.** We have now compared against 3DInfomax, GraphMVP, and DrugCLIP using published pretrained models (see our response to Reviewer CbPJ for the full table). All pretrained baselines underperform even ECFP on top-k shape retrieval recall, despite GraphMVP (R=0.37) and DrugCLIP (R=0.31) showing higher global correlation with shape scores than ECFP (R=0.20). This apparent discrepancy arises because higher global correlation does not translate to better top-k retrieval: ECFP succeeds at recovering top shape matches because molecules with the highest shape similarity often share 2D substructures (cf. Figure 4). SAND's large margin over all baselines (R=0.81–0.86) confirms that training directly on shape similarity is essential. The most directly comparable shape-trained methods (Atsango et al. 2022; Lzicar et al. 2024) did not publish code or are commercial-only. We emphasize our commitment to releasing code and weights, and will publish the training data as a benchmark.
>
> **Choice of GINE vs. Pretrained 3D-Aware Encoders.** We compared GINE against GPS and GATv2 (see our response to Reviewer CbPJ for the full table). GPS shows marginal improvements but is significantly larger and slower. Regarding pretrained 3D-aware encoders: as shown in our baseline results, pretrained 3DInfomax and GraphMVP representations do not help in the shape retrieval task, as they were pretrained on QM property prediction—an objective not aligned with 3D shape similarity. We note that SAND fundamentally requires storing a single embedding per molecule for scalable retrieval; 3D conformational information cannot be used directly at query time. As future work, 3D conformations could be incorporated during training (e.g., encoding multiple conformations) to teach the 2D encoder more shape-relevant features.
>
> **Training Data Bias.** The reviewer raises an interesting point. The distribution of training scores approximately follows N(μ=0.3, σ=0.19), so the model predominantly trains on moderate-similarity pairs—not just top hits. To systematically quantify the potential impact of pre-screening, we retrained SAND after dropping either all training pairs with score >0.5 or <0.0 (~15% of data each), mimicking two biases a pre-screening model could introduce. Recall@k of top-100 shape matches (k=10^x):
>
> | Dataset | x | All Scores | Scores < 0.5 | Scores > 0.0 |
> |---------|---|------------|--------------|--------------|
> | ChEMBL  | 2 | 0.34±0.15  | 0.33±0.15    | 0.32±0.16    |
> | ChEMBL  | 3 | 0.75±0.19  | 0.75±0.19    | 0.72±0.19    |
> | ChEMBL  | 4 | 0.97±0.07  | 0.97±0.07    | 0.96±0.07    |
> | Enamine | 2 | 0.50±0.10  | 0.50±0.10    | 0.49±0.10    |
> | Enamine | 3 | 0.89±0.08  | 0.89±0.08    | 0.88±0.08    |
> | Enamine | 4 | 0.99±0.01  | 0.99±0.01    | 0.99±0.01    |
>
> Performance remains largely unaffected, demonstrating that the pre-screening does likely not introduce harmful bias. Notably, removing all scores >0.5 has virtually no impact on top-100 recall, indicating that the model generalizes from learning shape comparisons in the moderate-similarity range (<0.5) to correctly ranking the highest-scoring matches. Exhaustive N×N scoring over the full library would be computationally prohibitive and would predominantly provide uninformative low-similarity pairs. We will include this analysis in the revision.
> Also note, our evaluation protocol is fully independent of SAND's training: test queries are compared against a fixed reference set (100k molecules) with exhaustive pairwise shape overlay—no SAND pre-screening is involved.
>
> We hope the new experiments and clarifications address the reviewer's concerns and
> kindly ask them to consider raising their score.

---

> > ### Author Rebuttal · Reviewer_95Jb · 2026-04-02
> >
> > My initial score is already quite high, so I’ve decided to keep my score.

---

> > > ### Author Response · Authors · 2026-04-07
> > >
> > > Thank you for the positive assessment and for confirming that your concerns have been addressed. We will incorporate all additions presented during the rebuttal in the final version.

---

### Official Review · Reviewer_f8LX · 2026-03-10

**Soundness:** 2
**Presentation:** 3
**Significance:** 2
**Originality:** 2
**Overall Recommendation:** 5
**Confidence:** 2

**Summary:**

This paper proposes SAND, a ligand-based virtual screening method that learns 2D graph embeddings whose cosine similarity approximates expensive 3D shape overlap, and couples this with end-to-end IVF-PQ quantization-aware training so the learned embeddings can be stored and searched efficiently in FAISS. The paper positions the method as a scalable replacement for exhaustive conformer-based shape screening on ultra-large libraries, with an additional application to retrieving purchasable analogs for generative chemistry outputs. Empirically, the paper reports strong correlation with ROSHAMBO2-derived shape scores (R=0.86 on Enamine and R=0.81 on ChEMBL), large gains over ECFP retrieval in Recall@k for top shape neighbors, improved compression relative to post-hoc FAISS quantization, and very fast search over a 10B-molecule Enamine index.

**Compliance With Llm Reviewing Policy:**

Affirmed.

**Final Justification:**

Thanks for the answers. I've raised my recommendation to accept from weak reject, but reduced the confidence, since I am still not sure that ECFP-based features are a strong baseline.

**Key Questions For Authors:**

1. The paper evaluates retrieval quality using ROSHAMBO2-derived labels, but compares against ROCS only in runtime. Can the authors provide a direct quality comparison against ROCS rankings on a shared benchmark?

2. The shape-label training set is built by scoring each query only against the top-2048 candidates from an earlier version of the same method. How much does this self-pre-screening bias the task? Can the authors evaluate with candidate pairs sampled independently of SAND?

3. The ROCS timing in Table 2 is not sufficiently specified. Are these numbers direct wall-clock measurements or throughput-based extrapolations? Please report the exact ROCS setup, hardware, conformer counts, and whether preprocessing costs were excluded symmetrically.

4. The representation-quality comparison is mainly against ECFP/Morgan fingerprints. Why are there no direct comparisons to prior learned 3D/shape-aware molecular descriptors discussed in related work?

5. In Section 4.4, the downstream validation compares against randomly sampled controls. Can the authors include a stronger ligand-based baseline (at least ECFP or a conventional shape-retrieval pipeline), rather than only random selection?

**Limitations:**

Yes

**Strengths And Weaknesses:**

Strengths:

The paper studies a relevant problem, namely, scalable ligand-based shape retrieval over very large libraries. The proposed combination of shape-aware representation learning and end-to-end IVF-PQ quantization is technically coherent and practically motivated, since it enables direct integration with FAISS. Empirically, the method improves over the post-hoc FAISS compression baseline in Figure 2 and over ECFP retrieval in Table 1.

Weaknesses:

For me, the quality of the embeddings is unclear since the baseline set is weak. The only baseline for quality of embeddings is ECFP / Morgan fingerprints. There are no comparisons with modern learned shape-aware embeddings. The authors compare SAND with ROCS (that is cited as a 1996 paper) only in terms of speed. Even FastROCS from OpenEye is not included. Also, in Table 2, ROCS timings are suspicious (10^11 seconds is given like something that was not extrapolated but measured).

Moreover, the practical validation in Section 4.4 is useful as a demo but not yet strong evidence of real-world hit expansion. It is limited to two kinase targets, compares against random controls rather than competing retrieval tools, and relies on docking/MM-PBSA-style proxies rather than measured activity or enrichment against known actives/decoys. I view this section as suggestive rather than conclusive.

---

> ### Author Rebuttal · Authors · 2026-03-31
>
> We thank the reviewer for their detailed feedback. We address each concern below.
>
> **ROCS Quality Comparison** We appreciate this important question.
> Actually, we initially trained SAND on ROCS scores and planned to release those weights. However,
> after consultation with OpenEye, we learned that their license prohibits publishing
> model weights trained on ROCS outputs. We therefore shifted to the open-source tool
> ROSHAMBO2. We note that ROSHAMBO2 and ROCS overlay scores do not correlate perfectly,
> primarily due to different color (pharmacophore) definitions, so evaluating a
> ROSHAMBO2-trained model against ROCS scores would not be meaningful—if ROCS-quality
> rankings are desired, one should train on ROCS labels directly. The model we trained on
> ROCS scores shows a similar trend for recall/compression as the ROSHAMBO2-based
> model, while achieving even higher correlation (Enamine R=0.91, ChEMBL
> R=0.83), likely due to the larger training set (3 million queries vs 200 thousand used
> for the ROSHAMBO2-based training). We are currently checking with OpenEye whether the labeled validation
> data can be shared; we will include this comparison in the appendix.
>
> **ROCS Timing Specification.** We clarify that the "ROCS" row in Table 2 refers to
> FastROCS (GPU); we apologize for the unclear labeling and will fix this. ROCS/FastROCS timings in
> Table 2 are indeed throughput-based extrapolations, not wall-clock measurements over 10B
> molecules. We will clearly state this in the revision and report the exact setup
> (hardware, conformer counts (50-300 based on rotatable bounds)). We will additionally add ROSHAMBO2 timings: on 8×H100,
> ROSHAMBO2 requires ~10^7 s (extrapolated) for 10B molecules—slower than FastROCS
> primarily due to I/O bottlenecks and less optimized file formats (as acknowledged by the ROSHAMBO2 authors).
> Regarding preprocessing costs: the reported ROCS/ROSHAMBO2 timings exclude conformer
> generation, which actually makes the comparison *more favorable* to these tools. Once the SAND index is built, no per-query
> preprocessing is required—retrieval is a single vector lookup. We will clarify this
> asymmetry in the revision.
>
> **Training Data Bias and Independent Evaluation.** We address this concern with ablation experiments in our response to Reviewer 95Jb, where we show that selectively removing high- or low-scoring training pairs has negligible impact on recall. We further emphasize that our test evaluation is already independent of SAND's training pipeline: test queries are compared against a fixed reference set (100k molecules) using exhaustive pairwise shape overlay—no pre-screening by SAND is involved.
>
> **Stronger Baselines.** We have added comparisons against 3DInfomax, GraphMVP, and DrugCLIP using published pretrained models (see our response to Reviewer CbPJ for the full table). All baselines underperform even ECFP on top-k shape recall, confirming that ECFP was already a competitive baseline for this task. SAND's advantage stems from being trained directly on the target shape similarity metric.
>
> **Section 4.4 Downstream Baseline.** We have strengthened the evaluation in two ways:
> (1) we now include ECFP-based (radius 2, 2048 bits) nearest-neighbor retrieval as a stronger
> ligand-based baseline in addition to random controls, and (2) all docking experiments
> were repeated with side-chain flexibility to improve scoring reliability.
> Across both targets, SAND-matched molecules consistently and substantially outperform both baselines. For JNK3, SAND achieves a mean Chemgauss4 score of −12.91 (median −13.04) ± 1.72, compared to −7.47 (−7.89) ± 2.77 for the ECFP baseline and −7.27 (−7.46) ± 2.50 for random controls. For P38, SAND yields −9.67 (−9.77) ± 1.25 versus −5.89 (−6.31) ± 3.13 (ECFP) and −3.51 (−3.79) ± 3.37 (random). These results confirm that the advantage of SAND-matched molecules is not merely due to generic chemical similarity but reflects the 3D shape-aware retrieval captured by the learned representations. We note that conventional shape-retrieval tools such as FastROCS would require month of GPU time to screen the same candidate pool.
> We acknowledge that the evaluation is limited to two kinase targets and relies on docking-based scoring rather than experimental binding data. However, we note that (a) prospective wet-lab validation is beyond the scope of this work, which introduces a representation learning method, and (b) physics-based docking scores remain a widely accepted computational proxy in the virtual screening literature. We believe the results are sufficient to demonstrate that SAND effectively retrieves molecules with meaningfully better predicted binding affinity than standard fingerprint-based retrieval, motivating future experimental follow-up. We have clarified this framing in the revised manuscript.
>
> We hope these additions address the reviewer's concerns and kindly ask them to consider raising their score.

---

> > ### Author Rebuttal · Reviewer_f8LX · 2026-04-05
> >
> > Thanks for the answers. I've raised my recommendation to accept from weak reject, but reduced the confidence, since I am still not sure that ECFP-based features are a strong baseline.

---

> > > ### Author Response · Authors · 2026-04-07
> > >
> > > Thank you for raising your score and for engaging thoughtfully with our rebuttal. Regarding the remaining baseline concern: we agree that ECFP features are not inherently a strong shape baseline. However, as discussed in detail in our response to Reviewer CbPJ, top-k shape retrieval is partly driven by shared substructures (of query and match compounds), which explains why ECFP performs surprisingly well in this specific setting — and actually outperforms 3DInfomax, GraphMVP, and DrugCLIP on top-100 recall. Conversely, when evaluating global correlation across the full range of shape scores, these 3D-aware methods do outperform ECFP, as expected. SAND substantially outperforms all baselines in both settings. We note that in the docking evaluation (Section 4.4), ECFP also serves as an appropriate complementary baseline precisely because it represents a different retrieval paradigm (2D substructure vs. 3D shape), isolating the contribution of shape-aware selection; a second shape-retrieval tool such as FastROCS would be expected to retrieve similar molecules and is computationally infeasible at the billion-compound scale (cf. Table 2). We will incorporate all additions into the final version of the paper. Thanks again for your comments and suggestions!

---

### Official Review · Reviewer_CbPJ · 2026-03-16

**Soundness:** 3
**Presentation:** 2
**Significance:** 3
**Originality:** 3
**Overall Recommendation:** 4
**Confidence:** 3

**Summary:**

This paper studeis the virtual screening problem for small molecules, which is typically computationally prohibitive. This work introduces SAND (Shape-Aware Neural Descriptor), a framework designed to enable rapid, billion-scale, shape-based virtual screening by learning to map 2D molecular graphs directly to a shape-aware vector space. The approach uses a differentiable Spearman correlation loss to enable rank-preserving contrastive learning and then use a two-level IVF-PQ discretization step to facilitate fast search with FAISS. The authors demonstrate that scalability of the proposed method searching over 10 billion molecules and showcase the application with deep generated models.

**Compliance With Llm Reviewing Policy:**

Affirmed.

**Final Justification:**

The author has addressed my concerns during rebuttal. Overall, the paper is well-written and sound, though the current version still needs revision. Therefore, my recommendation is weak accept.

**Key Questions For Authors:**

See above.

**Limitations:**

Yes

**Strengths And Weaknesses:**

Overall, this is a good paper addressing an important and challenging problem in virtual screening. The proposed SAND framework is technically sound and demonstrates promising empirical performance, particularly in its $10^8\times$ speedup over traditional methods like ROCS. The ability to rapidly identify commercially available chemical matters is a significant contribution that helps bridge the gap between deep generative model outputs and synthetically tractable chemical matter.

However, there are several major concerns:
1. The paper lacks many critical techincal details necessary for reproducibility and thorough evaluation:
- What are the detailed configuration for the encoder? Details such as the number of layers, hidden dimensions, and dropout rates are missing.
- What are the training hyperparameters? The paper does not specify essential hyperparameters, including the learning rate, batch size, optimizer settings, or the weight coefficients ($\gamma, \beta$) used in the final multi-objective loss function.
2. Several key architectural decisions are presented without sufficient justification:
- The authors utilize GIN, though there have been significant recent advancements in molecular representation learning. The choice of GINE should be justified through empirical comparison.
- The use of IVF-PQ for discretization is a core component of the method, but its superiority over other vector database indexing or quantization strategies is not proved.
- A formal ablation study is required to understand the contribution of specific components, such as the differentiable Spearman correlation loss versus standard contrastive losses.
3. The only baseline used for comparison is ECFP. However, the field of molecular representation learning is mature, with many existing methods (e.g., 3D-Infomax, GraphMVP, or SchNet, as presented by the authors) that could potentially be adapted for shape-aware retrieval. To demonstrate the true value of SAND, the authors should include comparisons against these more modern representation learning baselines.

I see clear potential in the proposed method; however, the lack of technical clarity and the lack of baseline comparisons prevent the paper from meeting the high standards of a top-tier machine learning conference. I am willing to increase my score if the authors can provide the missing technical details and address the concerns above.

Minor: Most references in the bibliography are missing publication years. Please ensure all citations are complete and follow standard academic formatting.

---

> ### Author Rebuttal · Authors · 2026-03-31
>
> We thank the reviewer for their constructive feedback and willingness to reconsider their score. We address each concern below.
>
> **Hyperparameters and Reproducibility.** The reviewer is right to flag this omission. A
> detailed hyperparameter table will be added to the appendix. Key settings: Encoder: GINE
> (3-layer MLP), hidden_dim=1024, num_layers=7, out_dim=512. Loss weights: γ=0.5, α=0.2,
> β_ivf=0.5, β_pq=0.25. Training: AdamW (wd=0.05), cosine annealing with linear warmup
> (LR=5e-4, min_LR=1e-5, warmup=10k, decay=1M iters), batch_size=1024, trained on 4×H100
> for ~1 week. Note also that for reproducibility, complete code, pretrained weights, and training data will be released upon
> acceptance.
>
> **Ablation Studies and Baseline Comparisons.** Based on the reviewer's comments, we conducted a systematic set of additional experiments to evaluate the impact of the encoder architecture, the loss function, and the choice of baselines. All experiments use the same evaluation protocol as Table 1 (Recall@k of top-100 shape matches, k=10^x). We consolidate the results below:
>
> | Dataset | x | SAND (GINE, spearman)| GPS| GATv2 | Triplet | MSE | ECFP | 3DInfomax| GraphMVP | DrugCLIP|
> |---------|---|---------------|-----------|-----------|-----------|-----------|-----------|-----------|-----------|-----------|
> | Enamine | 2 | 0.50±0.10 | 0.51±0.11 | 0.50±0.10 | 0.42±0.13 | 0.46±0.10 | 0.13±0.06 | 0.03±0.02 | 0.05±0.03 | 0.04±0.03 |
> | Enamine | 3 | 0.89±0.08 | 0.91±0.09 | 0.89±0.08 | 0.82±0.12 | 0.88±0.09 | 0.30±0.11 | 0.10±0.07 | 0.16±0.10 | 0.14±0.10 |
> | Enamine | 4 | 0.99±0.01 | 0.99±0.01 | 0.99±0.01 | 0.98±0.02 | 0.99±0.01 | 0.60±0.13 | 0.34±0.13 | 0.48±0.17 | 0.46±0.17 |
> | ChEMBL  | 2 | 0.34±0.15 | 0.39±0.11 | 0.33±0.12 | 0.28±0.11 | 0.31±0.12 | 0.05±0.04 | 0.01±0.01 | 0.02±0.02 | 0.02±0.03 |
> | ChEMBL  | 3 | 0.75±0.19 | 0.81±0.10 | 0.74±0.11 | 0.66±0.09 | 0.71±0.12 | 0.15±0.09 | 0.04±0.04 | 0.10±0.07 | 0.10±0.08 |
> | ChEMBL  | 4 | 0.97±0.07 | 0.98±0.07 | 0.97±0.08 | 0.93±0.07 | 0.95±0.09 | 0.42±0.14 | 0.21±0.11 | 0.42±0.17 | 0.36±0.17 |
>
> *Encoder (cols 3–5):* GPS (Rampášek et al. 2022) and GATv2 (Brody et al. 2021) use the same Spearman loss, substituting only the
> GNN backbone. GPS shows marginal gains, likely due to its 3× larger capacity (87M vs 28M
> parameters for GINE/GATv2). Importantly, SAND is encoder-agnostic: its core
> contributions—the differentiable Spearman loss and end-to-end quantization-aware
> training—are independent of the backbone. Any GNN can be substituted. In fact, the
> training data we plan to release might provide a useful benchmark for GNN architecture.
>
> *Loss function (cols 6–7):* Triplet and MSE variants use the same GINE encoder, replacing only the loss. The Spearman loss consistently outperforms both: because ground-truth overlay scores are comparable across samples, it leverages batch-wide ranking information, providing richer signal than per-sample triplet loss. Unlike MSE, it directly optimizes for ranking rather than absolute score regression.
>
> *Baselines (cols 8–11):* 3DInfomax and GraphMVP both use contrastive learning between 2D
> graph and 3D conformer views to inject geometric awareness, pretrained on QM property
> datasets (QMUGS/GEOMDrugs). DrugCLIP uses contrastive learning to match molecule
> embeddings to protein pocket embeddings, which may encode shape bias.
> Interestingly, GraphMVP (R=0.37) and DrugCLIP (R=0.31) achieve higher global correlation
> with shape scores than ECFP (R=0.20), yet all underperform ECFP on top-100 recall. This
> is probably because the top shape matches often share common substructures with the query,
> making ECFP a surprisingly competitive baseline for top-k retrieval despite its lack of
> explicit 3D awareness. However, all baselines remain far below SAND (R=0.86), which
> is trained directly on shape overlay scores. We note that SchNet, requires 3D coordinates as input and cannot produce embeddings from 2D graphs, making it inapplicable for this 2D-based retrieval setting.
>
> **Quantization Alternatives.** We additionally compared our end-to-end IVF-PQ against Residual Vector Quantization (RVQ), trained with the same framework. RVQ achieves superior overall correlation (Enamine R=0.84 vs 0.78; ChEMBL R=0.81 vs 0.73), consistent with RVQ's known advantage in reconstruction accuracy (Babenko & Lempitsky, CVPR 2014). However, Recall@k for top-100 shape matches remains comparable across both methods. Crucially, RVQ's iterative beam-search encoding makes it significantly slower at query time in FAISS compared to IVF-PQ, which benefits from highly optimized GPU kernels and precomputed look-up tables (Douze et al., 2024). For billion-scale retrieval where query latency is critical, IVF-PQ therefore remains the more practical choice. We will include this discussion of the accuracy–speed tradeoff in the revision.
>
> **Bibliography.** We will fix the missing publication years.
>
> We hope these additions address the reviewer's concerns and kindly ask them to consider raising their score.

---

> > ### Author Rebuttal · Reviewer_CbPJ · 2026-04-01
> >
> > Thanks for the detailed response. I think most of my concerns are resolved. The final version of the paper should include the revisions made during rebuttal.

---

> > > ### Author Response · Authors · 2026-04-07
> > >
> > > Thank you for the positive reassessment and for engaging with our rebuttal. We are glad that our additional experiments and clarifications addressed your concerns. We confirm that all additions and analyses presented during the rebuttal — including the hyperparameter table, ablation studies, expanded baselines, quantization comparison, and bibliography fixes — will be incorporated into the final version of the paper. Thank you for your suggestions — we believe they have genuinely strengthened the manuscript.

---

### Decision · Program_Chairs · 2026-04-30

**Decision:**

Accept (regular)

**Comment:**

The paper presents SAND, a method that learns shape-aware molecular embeddings directly from 2D graphs so that shape-based virtual screening can be turned into large-scale vector retrieval. It combines a rank-preserving differentiable Spearman loss with a two-level IVF-PQ discretization step to facilitate fast search. Experiments show that SAND enables billion-scale retrieval while maintaining strong correlation with expensive 3D shape-overlap scores.

The paper was well received, especially after the rebuttal. The reviewers appreciated the strong underlying motivation to solve a practical bottleneck in virtual screening, and the coherent architecture combining shape-aware learning with fast and scalable retrieval. Weaknesses were identified especially about missing details on reproducibility, and missing stronger baselines. Both seem to have been addressed during the rebuttal.

I agree with the reviewers that this paper must be intended more as a successful integration of different tools, rather than a methodological breakthrough. Nonetheless, the discussion clearly indicates that this paper's contributions are solid and deliver high value to the community. I recommend acceptance.